# Early-life exposure to Ivermectin alters long-term growth and disease susceptibility

**Taegan A. McMahon\***, **Shannon Fernandez-Denmark, Jeffrey M. Grim** ⬥ *

Department of Biology, University of Tampa, Tampa, FL, United States of America

* taeganmcmahon@gmail.com (TAM); jgrim@ut.edu (JMG)

**Data Availability Statement:** All relevant data are within the paper and its Supporting Information files.

**Funding:** Funding was provided to TAM (IOS award- 1754862) by the United States National

## Abstract

Ivermectin is a broad-spectrum antiparasitic medicine, which is often used as a treatment for parasites or as a prophylaxis. While studies have looked at the long-term effects of Ivermectin on helminths, studies have not considered the long-term impacts of this treatment on host health or disease susceptibility. Here, we tracked the effects of early life Ivermectin treatment in Cuban tree frogs (*Osteopilus septentrionalis*) on growth rates, mortality, metabolically expensive organ size, and susceptibility to *Batrachochytrium dendrobatidis* (Bd) infection. One year after exposure, there was no effect of Ivermectin exposure on frog mass ($X^2_1$ = 0.904, p = 0.34), but when tracked through the exponential growth phase (~2.5 years) the Ivermectin exposed individuals had lower growth rates and were ultimately smaller ($X^2_1$ = 7.78, $p$ = 0.005; $X^2_1$ = 5.36, $p$ = 0.02, respectively). These results indicate that early life exposure is likely to have unintended impacts on organismal growth and potentially reproductive fitness. Additionally, we exposed frogs to Bd, a pathogenic fungus that has decimated amphibian populations globally, and found early life exposure to Ivermectin decreased disease susceptibility (disease load: $X^2_1$ = 17.57, $p$ = 0.0002) and prevalence (control: 55%; Ivermectin: 22%) over 2 years after exposure. More research is needed to understand the underlying mechanism behind this phenomenon. Given that Ivermectin exposure altered disease susceptibility, proper controls should be implemented when utilizing this drug as an antiparasitic treatment in research studies.

## Introduction

Parasitic nematodes are cosmopolitan and infect an extreme variety of taxonomic groups (e.g. beetles [1,2], amphibians [3] and humans [4]). In humans alone parasitic soil helminths are estimated to infect more than 1.45 billion people globally [4], many of whom are at risk of increased morbidity or mortality. To this end, Ivermectin is often used as a long-term anthelmintic and is broadly effective against many different nematode infections (e.g. *Ascaris lumbricoides*- the most common human helminth infection, *Necator americanus*, *Ancylostoma duodenale* [4] and *Onchocerca vulvulas* [2]). In most cases, the health cost of taking Ivermectin is thought to be better than the alternative infection. Ivermectin is often only effective against the juvenile stages (microfilariae; [5]), which means early treatment and long-term

Science Foundation (https://www.nsf.gov/div/index.jsp?div=IOS), and the Faculty Development Dana and/or Delo Grants and Biology Department Student Research Funds were provided to JMG and TAM by the University of Tampa (www.ut.edu). The funders had no role in study design, data collection and analysis, decision to publish, or preparation of the manuscript.

**Competing interests:** The authors have declared that no competing interests exist.

administration of this drug in at-risk populations, (*e.g.* pre-pubescent children) is necessary to reduce the impact of infections [2].

Ivermectin is an incredibly important medication used in agriculture and pet trade, as well. For example, it is used to prevent *Dirofilaia immitis*, canine heartworm [6,7], to clear lung-worm infections in US cattle [8], and it is one of the most common anthelmintic medications used in sheep [9] and equines in the UK [10]. The adult nematodes can live for 5–10 years, and while Ivermectin does not kill the adults, it can be used reduce local transmission events by killing the microfilariae. Again, in both the agricultural system and with our own pets, we often utilize early life administration of Ivermectin to try to control nematode infections.

Amphibians, arguably the most at-risk taxonomic group in terms of declines and extinctions [11], are another group known to suffer from severe helminth infections [3]. Amphibians are declining globally and are often collected from the wild to be cared for by aquariums and zoos. In order to provide the highest chance of survival and to reduce the chance of parasitic transmission to the other captive individuals, facilities clear amphibians of their helminth infections with Ivermectin upon arrival [12]. Many of these species are threatened or endangered in the wild, and so the survival of these captive individuals is crucial.

Ivermectin functions by altering the glutamate-gated chloride channel receptors [5] and causing nematode paralysis. It is important to note, however, that while these channels are found in the peripheral neuromuscular systems of invertebrates, they are also found in the central nervous system of vertebrates [13,14]. Consequently, this commonly used anthelmintic may be targeting invertebrates, while impacting the vertebrate host as well.

Here, we assessed the impacts of early life exposure of Ivermectin on long term growth, health and disease susceptibility in amphibians. Early life exposure to pesticides is known to have long-lasting and often unpredictable impacts on amphibians, including increased mortality and disease risk into adulthood [15]. Therefore, it seems plausible that Ivermectin exposure may similarly impact health. Amphibians are one of the groups often affected by Ivermectin exposure, and importantly, they share approximately 1700 genes with human disease associations [16]. Additionally, they are a useful model for exploring the impacts of pesticides like Ivermectin on humans because there is overlap between the two immune and endocrine systems [17]. Understanding the impacts of chemical exposure on amphibians themselves is important, and they are an underutilized model for studying the chemical exposure on human health.

## Results

There was no effect of Ivermectin treatment on mortality ($X^2_1$ = 0.904, p = 0.34; survival rates of 45% and 55% in control and Ivermectin-dosed groups, respectively) and there was no difference in frog mass at one-year post exposure (average mass ± SEM: control: 0.668 ± 0.02; Ivermectin: 0.674 ± 0.03; $X^2_1$ = 0.02, $p$ = 0.877; Fig 1). Individuals exposed to Ivermectin had reduced growth rate (2.5 years of growth: $X^2_1$ = 7.78, $p$ = 0.005; Fig 2) and lower change in mass ($X^2_1$ = 5.36, $p$ = 0.02) at the end of the experiment, relative to unexposed animals. There was no significant effect of Ivermectin treatment on metabolically expensive organ mass (liver: $X^2_1$ = 0.006, p = 0.93, heart: $X^2_1$ = 0.730, p = 0.39, or spleen: $X^2_1$ = 0.21, p = 0.90), and independent of treatment, the size of the frog was a significant correlate for liver and heart but not spleen (liver: $X^2_1$ = 5.33, $p$ = 0.02, heart: $X^2_1$ = 5.03, $p$ = 0.02, spleen: $X^2_1$ = 0.03, $p$ = 0.86).

Frogs with early life Ivermectin exposure had decreased Bd prevalence (control: 55% and Ivermectin: 22%) and disease load ($X^2_1$ = 17.57, $p$ = 0.0002; Fig 3). We also found spleen mass and Bd load correlated ($X^2_1$ = 6.60, $p$ = 0.01), but that there was an interactive effect of Ivermectin exposure and spleen mass on Bd load ($X^2_1$ = 11.60, $p$ = 0.0007; Fig 4). There was a

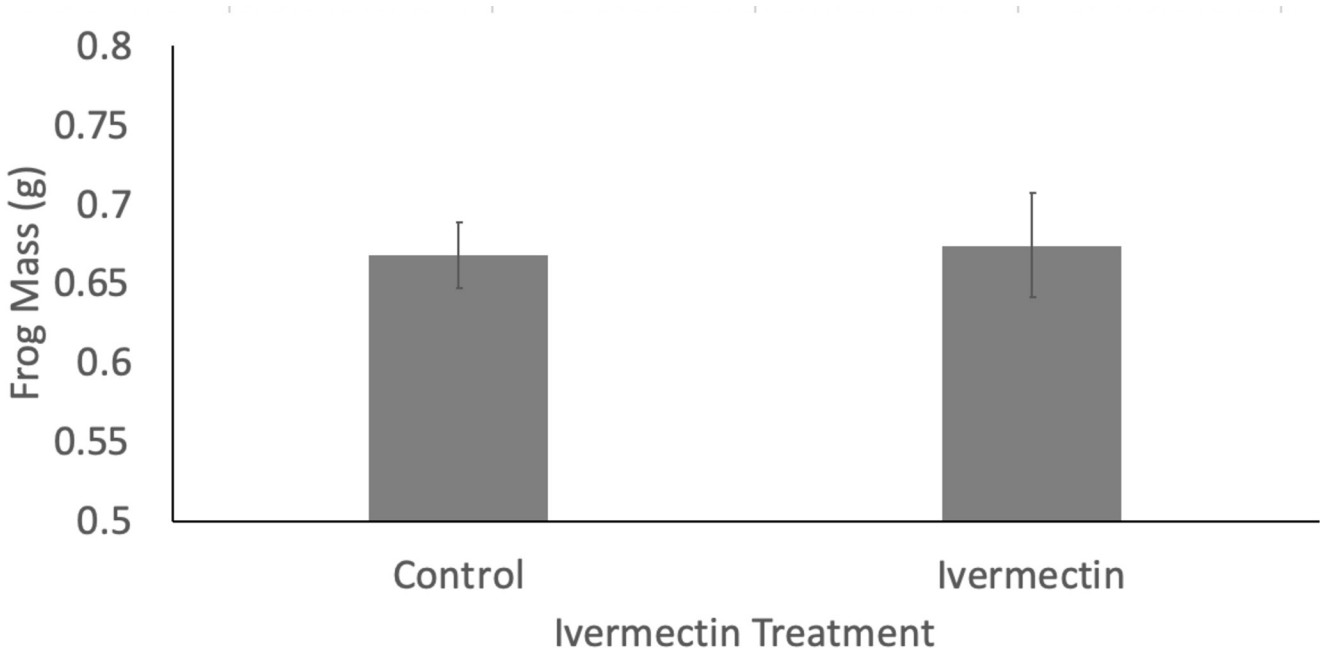

**Fig 1. The effects of early life exposure of Ivermectin on mass of Cuban tree frogs (*Osteopilus septentrionalis*) one year after metamorphosis.** No significant difference in mass was observed. Shown are means ± SEM.

negative correlation between spleen size and Bd load for individuals exposed to Ivermectin, while for the control individuals the correlation was positive (Fig 4).

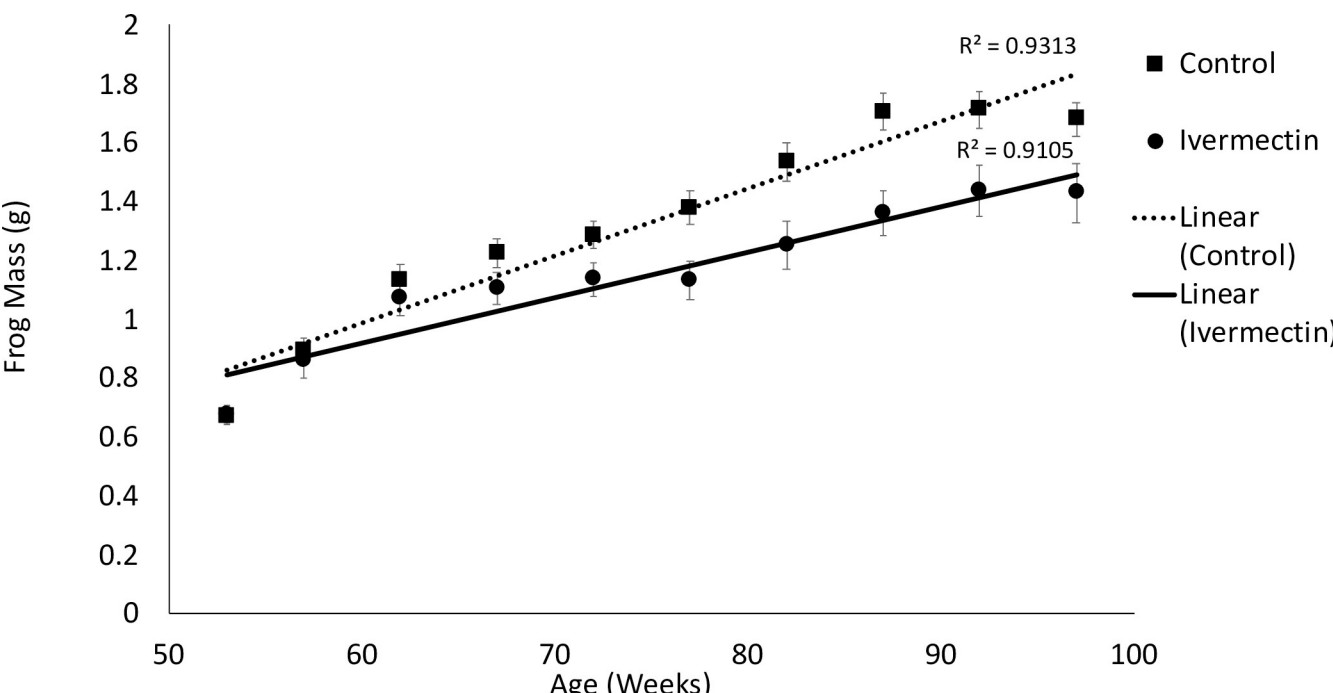

**Fig 2. The effects of early life exposure of Ivermectin on the long-term growth of Cuban tree frogs (*Osteopilus septentrionalis*).** Individuals exposed to Ivermectin had slower growth rates and weighed less, relative to controls. Shown are means ± SEM.

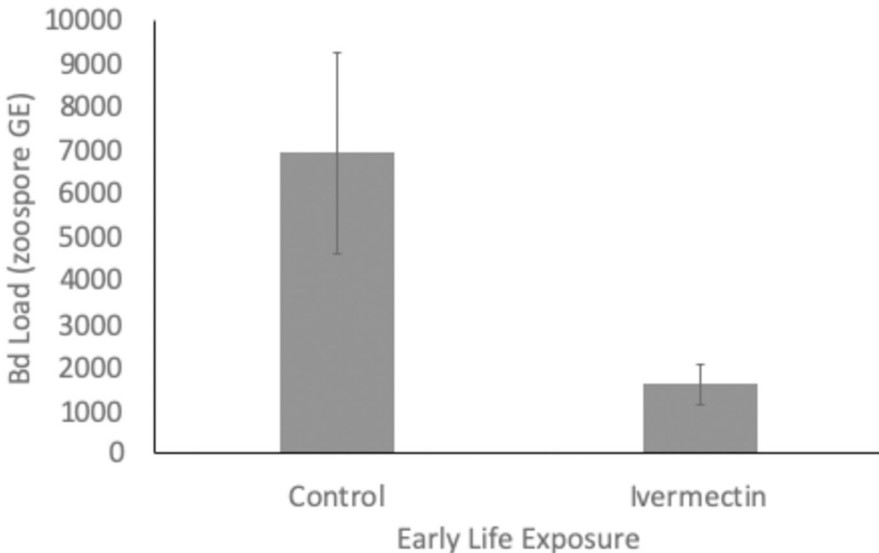

**Fig 3. The effects of early life exposure of Ivermectin on the disease susceptibility in Cuban tree frogs (*Osteopilus septentrionalis*) to *Batrachochytrium dendrobatidis* (Bd; Bd load is reported in zoospore genome equivalents).** Individuals exposed to Ivermectin were less susceptible to Bd infection compared to controls. Shown are means ± SEM.

## Discussion

Ivermectin is an extremely common antiparasitic medication used in humans and other vertebrates both prophylactically and in treatment regimes. Studies have assessed the mortality and short-term risk associated with administration of this drug, and generally the short-term side effects appear minimal especially when compared to the benefit of the drug [5,18,19]. Our results support this in that metamorphic amphibians exposed at an early developmental stage did not

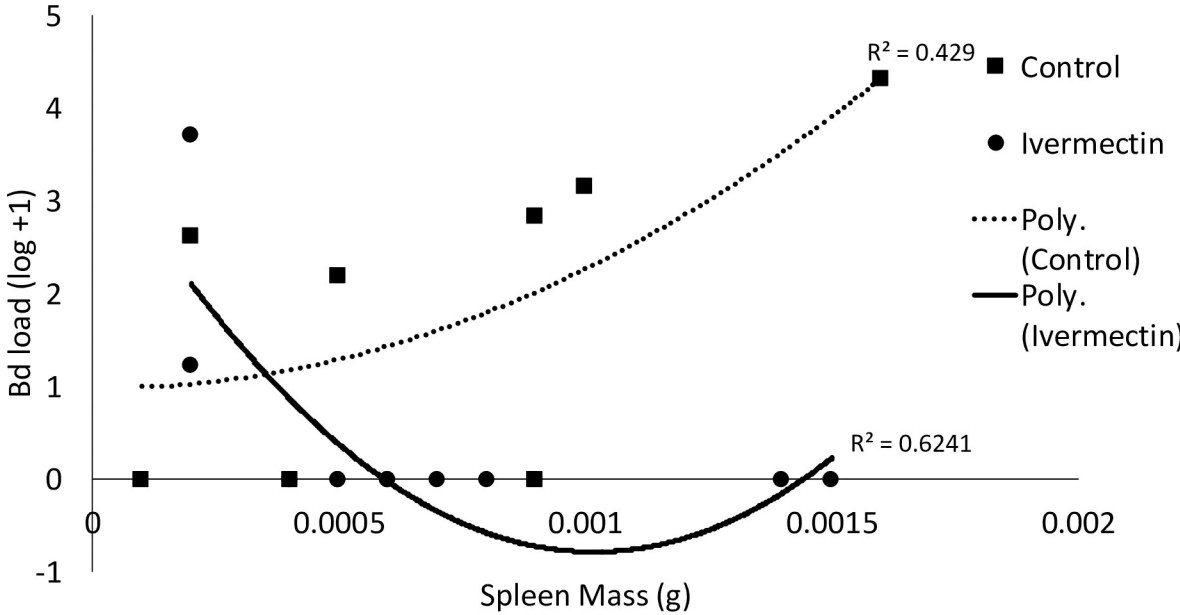

**Fig 4. There was an interaction between Ivermectin exposure and spleen size and their impact on *Batrachochytrium dendrobatidis* (Bd) load in Cuban tree frogs (*Osteopilus septentrionalis*).**

have altered growth up to one year after exposure (Fig 1). This is consistent with other studies that found short-term consequences were mild even with repeated or stronger than FDA recommended doses [18,19], however we do report long-terms risks of early Ivermectin exposure.

Cuban tree frogs undergo sexual maturity at approximately 17 and 36 weeks for males and females, respectively (reviewed by [20]), but they continue to grow for several years. Here, we exposed adolescent Cuban tree frogs (~ 4 weeks post metamorphosis) to Ivermectin and tracked their growth through the critical growth period (approximately 2.5 years). Despite no evidence of growth differences 1 year after exposure, we found that individuals exposed to Ivermectin, just once as young developing metamorphs, had reduced growth late into their critical growth period (Fig 2). This puts the Ivermectin exposed individuals at a disadvantage long-term, as larger males and females are reproductively more fit [21]. Given that these individuals only experienced one dose, it is likely that our results underestimate the true impact of repeated Ivermectin exposure during early development.

Organisms operate with limited energetic resources, and during development these resources must be differnetially allocated to growth, the development of crucial organs (e.g. heart and liver), and pathogen defense (e.g. immune response). We found that while there was a positive correlation between frog mass and organ (liver and heart) mass, which was expected, there was no effect of Ivermectin treatment on the mass of these metabolically expensive organs. Ivermectin exposure reduced whole body growth and growth rate when compared to control animals, but did not result in smaller hearts, livers, or spleens of these individuals. This indicates Ivermectin influences growth in organ-specific ways, as metabolically expensive and important organs likely received an increased allocation of resources relative to whole body growth following Ivermectin treatment. This disparate allocation is not surprising given that studies have shown that resources are shunted toward critical organs and tissues in vertebrates in tissue-specific ways and under resource-limiting conditions [22,23].

We found that individuals exposed to Ivermectin were less susceptible to infection with the pathogenic fungus Bd (Figs 3 and 4). Thus, Ivermectin treatment may actually provide a secondary protection against other infectious agents, consistent with the the ongoing discovery of new therapeutic applications of the drug (reviewed in [5]) and despite the fact that Bd itself is immunosuppressive [24–26]. Interestingly, there was an interaction between spleen size and Ivermectin exposure in relation to Bd load. For individuals exposed to Ivermectin, there was a negative correlation between spleen size and Bd load and we saw the opposite for control individuals (Fig 4). Interestingly, Bd exposure alone was found to have no impact on spleen size [27,28], but frogs with an acquired immune response to Bd had increased splenocyte production [29]. While further research is needed to fully understand this dynamic, we hypothesize that the absence of an existing internal parasite load following Ivermectin treatment impacts immune system allocation in response to new infections. Potentially, this permits a larger proportion of the immune response to be directed against the new pathogen, Bd. Given that Bd infection is a major causitive agent in the extreme extirpations and extinctions of amphibian species worldwide [11], reduction in disease load and prevalence is certainly an unexpected benefit. We caution against the use of this antiparasitic drug in treatment of Bd, as it has not been tested for this, and given that we did not eliminate the infection it is unlikely to be effective for that purpose. Future work should explore whether the secondary protection afforded by Ivermectin treatment extends to other species and other infectious agents. Consequently, administration of Ivermectin might have unintented positive clinical outcomes, despite evidence indicating resistance development in helminth populations [5].

Our results indicate just a single dose of Ivermectin during early-life can negatively impact growth rates over time, and therefore potentially reproductive fitness. It is important to note that if we had stopped tracking growth at one year, less than halfway through the frogs' critical

growth period, we would not have gotten a view of the long-term impacts of Ivermectin treatment. While we acknowledge the restraints of lab resources and time, our data indicate the time frame considered for short and long term studies should be evaluated in the context of life history stage and overall lifespan of the organism, as well as species-specific drug half-lives.

The clinical and pharmacological side effects of drug adminstration must be outweighed by the positive clinical outcomes. Ivermectin certainly meets that criterion given the reported significant reduction in microfilarial dermal loads after treatment and minimal side effects [reviewed by 5,30]. More recently, research has shown the potential efficacy of Ivermectin as therapuetic for SARS-CoV-2 *in vitro* [31], as a potential complimentary pharmaceutical treatment for combating malaria [32,33], and a recent meta-analysis provides promising evidence supporting the removal of the contraindications for children <15 kg [34]. However a growing literature suggests neonatal and early-childhood antibiotics impact child growth, BMI, and gut microbome in age-at-dose and sex-specific ways (*e.g.*, [35]), indicating early pharmaceutical interventions can have long-term implications on organism growth. While we are certainly not advocating that Ivermectin use be eliminated, we want to make sure that it is clear that our results indicate there may similarly be long-term impacts of Ivermectin exposure in early developmental stages. Additionally, given that Ivermectin exposure altered both growth and disease susceptiblity in the long-term, the use of proper controls in scientific studies examining or utilizing Ivermectin is crucial [32,33]. Without these controls the impact of Ivermectin exposure may lead to erroneous experimental results.

## Methods

### Animal collection and maintenance

Eggs were collected from invasive, wild Cuban tree frogs (*Osteopilus septentrionalis*) by deploying plastic breeding refugia in local Tampa, FL, USA habitats (27.9506° N, 82.4572° W). No permits were required for egg collection as Cuban tree frogs are an invasive species in FL and all collections were done on private land with owner's permission. No additional collecting permits were required. Briefly, frogs deposited eggs into these refugia overnight, and the eggs were collected and brought to the lab during the morning hours. Eggs were maintained in artificial spring water (ASW; [36]) with weekly water changes and reared through metamorphosis. Animals were never exposed to soil and therefore should not have acquired any nematode infections. Tadpoles were fed organic spinach *ad libitum*. Metamorphic frogs (~ 4 weeks post metamorphosis) were divided evenly and randomly between control and Ivermectin-dosed groups (n = 20/treatment). They were all given a single dose orally of either ASW (control) or 1% Ivermectin (2 mg/kg body mass/treatment) at the same time. All animals were transferred to individual 1 L plastic containers with dampened paper towels and fed calcium dusted crickets *ad libitum*. Mortality was checked daily, container changes occurred weekly, and the frogs were weighed every week through their exponential growth phase (approximately 2.5 years), including during the final two week Bd infection period for both control and Ivermectin exposed groups. Throughout this process all amphibians were maintained at approximately 23°C.

### Bd culture preparation

We cultured *Batrachochytrium dendrobatidis* (Bd; strain JEL 419 isolated in Panamá) on 1% tryptone agar plates at 18°C for two weeks. We flooded these Bd positive (Bd+) plates with ASW to suspend the infectious zoospores, and this liquid was homogenized from all the Bd + plates. We quantified the zoospore concentration with a hemocytometer and the Bd+ stock was diluted with ASW to $1.3\text{x}10^5$ zoospores/mL.

## Bd exposure and tissue sampling

To evaluate the effects of Ivermectin exposure on Bd susceptibility, all surviving adult frogs in both control (n = 9) and Ivermectin-exposed (n = 11) groups were dosed with 1 mL of the Bd + stock during the final two weeks of the overall 2.5 year experimental period. Bd+ stock was squirted directly onto the dorsal surface of each frog. The extra liquid was allowed to collect on the damp paper towels in the 1 L plastic container to promote infection establishment. Following the singular Bd+ dose, the frogs experienced a two-week infection period, during which container changes occurred weekly, they were fed calcium dusted crickets *ad libitum*, and they were maintained at 18°C to promote Bd growth. Animals were euthanatized immediately at the end of this two-week infection period with an overdose of topical benzocaine and they were pithed post mortem. A ventral skin sample was collected from each frog (1 cm$^2$) for quantitative PCR to determine Bd infection load (See [37] for the qPCR protocol used). In order to assess the potential effects of Ivermectin treatment on the partitioning of resources between whole animal growth and the growth of individual organs, we also sampled the metabolically-expensive heart, liver, and spleen from each euthanized individual. Organs were collected and rinsed in a 0.7% saline bath, and were weighed immediately. All animal collections and procedures were approved in writing by the University of Tampa Institutional Animal Care and Use Committee (IACUC# 2018–12).

## Frog health monitoring

All animals were monitored daily for health and mortality status. Any animal that appeared lethargic, was not alert upon inspection and did not appear to be healthy in any way were further examined. If an animal appeared unhealthy, they were placed on their back and if righting was not observed in 20 seconds (a conservative timeframe selected to reduce unnecessary discomfort) then the animal was considered moribund. At this point the animal would be euthanized with an overdose of topical benzocaine and they were pithed post mortem as an extra precaution. All animals that died during this experiment died quickly and naturally and did not experience an undue, prolonged unhealthy state; indeed, no animals were euthanized due to an unhealthy state. This was a long-term experiment lasting several years, and early life mortality is not unusual for amphibians. The research staff were trained directly by the chair of the University's IACUC committee.

## Statistical analyses

All statistical analyses were conducted in R statistical software [38]. We used a Cox-proportional hazards regression (package: survival, function: coxph) to determine if there was an effect of Ivermectin treatment on survival. We used a general linear model (package: stats; function: glm) to determine if there was an effect of treatment on: mass size at one year, growth rate, frog mass change, and organ size at the end of the experiment. We used a zero-inflated negative binomial Poisson regression to determine if treatment, spleen size or treatment*spleen size had an effect on Bd load (package: pscl, function: zeroinfl).

## Supporting information

**S1 Checklist.**
(DOCX)

**S1 Raw data. These are the raw data sets used for all statistical analyses and figures.**
(XLSX)

## Acknowledgments

We would like to acknowledge C. Nordheim and N. Laggan for their research assistance.

## Author Contributions

**Conceptualization:** Taegan A. McMahon, Jeffrey M. Grim.

**Data curation:** Taegan A. McMahon, Jeffrey M. Grim.

**Formal analysis:** Taegan A. McMahon.

**Funding acquisition:** Taegan A. McMahon, Jeffrey M. Grim.

**Investigation:** Taegan A. McMahon, Shannon Fernandez-Denmark, Jeffrey M. Grim.

**Methodology:** Taegan A. McMahon, Shannon Fernandez-Denmark, Jeffrey M. Grim.

**Project administration:** Taegan A. McMahon, Jeffrey M. Grim.

**Resources:** Taegan A. McMahon, Jeffrey M. Grim.

**Supervision:** Taegan A. McMahon.

**Writing – original draft:** Taegan A. McMahon, Shannon Fernandez-Denmark, Jeffrey M. Grim.

**Writing – review & editing:** Taegan A. McMahon, Shannon Fernandez-Denmark, Jeffrey M. Grim.

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
