## [Decision Letter · Decision Letter 0]

18 Jun 2021

PONE-D-20-33880

Early-life exposure to Ivermectin alters long-term growth and disease susceptibility

PLOS ONE

Dear Dr. Grim,

Thank you for submitting your manuscript to PLOS ONE. After careful consideration, we feel that it has merit but does not fully meet PLOS ONE’s publication criteria as it currently stands. Therefore, we invite you to submit a revised version of the manuscript that addresses the points raised during the review process.

Thank you for your patience. This review took much longer than I had hoped. It has now been fully reviewed. Both reviewers were quite positive about the quality of the writing and the results applied here. The reviewers did point out some concerns that need to be addressed prior to publication, including clarification on experimental methods and textual suggestions.

We look forward to receiving your revised manuscript.

Kind regards,

Adler R. Dillman, Ph.D.

Academic Editor

PLOS ONE

Additional Editor Comments (if provided):

Thank you for your patience. This review took much longer than I had hoped. It has now been fully reviewed. Both reviewers were quite positive about the quality of the writing and the results applied here. The reviewers did point out some concerns that need to be addressed prior to publication, including clarification on experimental methods and textual suggestions.

Journal Requirements:

https://journals.plos.org/plosone/s/file?id=ba62/PLOSOne_formatting_sample_title_authors_affiliations.pdf
/

3. In your Methods section, please provide additional location information of the collection sites, including geographic coordinates for the data set if available. 

4. In your Methods section, please provide additional information regarding the permits you obtained for the work. Please ensure you have included the full name of the authority that approved the collection sites access and, if no permits were required, a brief statement explaining why. 

We would like to acknowledge C. Nordheim and N. Laggan for their research assistance. Funding was provided by the National Science Foundation [IOS- 1754862 (TAM)] and the University of Tampa [Faculty Development Dana and Delo Grants (JMG and TAM) and Biology Student Research Funds (JMG and TAM)].

Funding was provided by the National Science Foundation [IOS- 1754862 (TAM)] and the University of Tampa [Faculty Development Dana and Delo Grants (JMG and TAM) and Biology Student Research Funds (JMG and TAM)].

Reviewers' comments:

Reviewer's Responses to Questions

**Comments to the Author**

1. Is the manuscript technically sound, and do the data support the conclusions?

Reviewer #1: Yes

Reviewer #2: Partly

2. Has the statistical analysis been performed appropriately and rigorously? 

Reviewer #1: Yes

Reviewer #2: Yes

3. Have the authors made all data underlying the findings in their manuscript fully available?

Reviewer #1: Yes

Reviewer #2: Yes

4. Is the manuscript presented in an intelligible fashion and written in standard English?

Reviewer #1: Yes

Reviewer #2: Yes

5. Review Comments to the Author

Reviewer #1: This is a well-written manuscript presenting the results of a long-term study on the effects of Ivermectin use in an amphibian species. The results are clearly presented with appropriate statistical analyses. R2 values should be added to Fig. 4. The role of the spleen in Bd infection dynamics is interesting. While the spleen size did not scale with body size, unlike the other organs (heart, liver), spleen size was correlated with Bd load. The Figure four results showing an interaction between Ivermectin exposure and spleen size and their impact on Batrachochytrium dendrobatidis (Bd) are fascinating, but the mechanism remains unclear. The authors may want to discuss additional studies to help clarify these findings, such as monitoring frogs post-treatment for Ivermectin half-life and clearance from the hosts. There are interesting associations with antibiotic use and growth and body mass index in humans during development. This may also be worthwhile to include in the discussion (e.g., doi: 10.1038/s41467-020-20495-4.). I personally suspect that the drug may have altered the microbiome of the amphibians which subsequently impacted immune function or directly impacts Bd resistance. Overall, this is an exciting study that has very careful conclusions that do not over-reach. The authors should be commended for their persistence at carrying out a long laboratory study looking for long-term effects.

Reviewer #2: General comments:

In the manuscript “Early-life exposure to Ivermectin alters long-term growth and disease susceptibility”, the authors assess the effects of a single exposure to Ivermectin on amphibian growth ad susceptibility to the pathogenic fungus Batrachochytrium dendrobatidis (Bd). Although I think these results warrant eventual publication, I think the manuscript needs to be improved in two main ways (in addition to minor adjustments elsewhere).

First, throughout the introduction and discussion, the authors highlight the potential application of studying long-term effects of Ivermectin exposure in amphibians to those effects in humans. Without heavy citation, I think that this is a leap. I think the authors’ efforts would be better directed towards focusing on the effects of Ivermectin on amphibians (which they do – I’d just like to see this be highlighted to a greater extent than a potential link to humans).

Second, the experimental design and timeline require clarification. I highlight in line-by-line comments below specific areas of confusion that I had while reading. I assessed the results and discussion based on what I thought the authors did, but I’d like to see if I should revise my assessment once the methods are clarified.

Specific comments:

L87 – I think what you’re trying to get at here is that exposure to other chemicals, such as pesticides, can have long-lasting effects in amphibians, and, therefore, it is possible for Ivermectin to have long-lasting effects as well, but you could make this connection clearer.

L91-94 – I don’t think you need to push the potential connection to human health as hard as you do here and throughout the manuscript. As you indicate, amphibians are important in their own right.

L97 – Please give some indication of survival rates in the different treatments.

L99 (and elsewhere) – Sometimes you capitalize “Ivermectin” and sometimes you don’t. Be consistent.

L101 – Specify that this result is relative to those not exposed to Ivermectin.

L102 – I think this is the first time you mention “metabolically-expensive organs”. It would be beneficial to explain sooner why you chose to measure the liver, heart, and spleen.

L108-109, and 118-120 – Indicate what error bars represent in all figure captions.

Figure captions (general) – I think it would be helpful to have all captions highlight a specific result, as the caption for Figure 4 does (there it’s the interactive effect of Ivermectin exposure and spleen size on Bd load).

L115-117 – Run-on sentence.

L120 – Should say “zoospore genome equivalents).”

L128-130 – Needs citation(s).

L141-144 – I think it’s a pretty big leap from metamorphic frogs to human children here, particularly without citation.

L149 – Specify Ivermectin treatment.

L149-152 – I’m having a hard time following this result. Is there a way to show it in figure form?

L150 – “Metabolically” is misspelled.

L159-160 – I think you’d benefit here from including results of prior studies regarding the relationship between spleen size and Bd load.

L178 – Remove “whole heartedly”.

L186 – “Pharmaceutical” is misspelled.

L191 – Remove the comma between “Ivermectin” and “is”.

L203 – Please explain the Ivermectin treatment in more detail. I see in L139 that there was a single Ivermectin exposure, but that should also be stated here. Additionally, were all frogs dosed with Ivermectin at the same absolute time (as opposed to the same amount of time between metamorphosis and exposure for each frog)? And if so, approximately how long after metamorphosis?

L218 – Relative to Ivermectin exposure, when were frogs exposed to Bd? After reading through a couple times, I’m guessing this was in the final two weeks of the experiment? And If so, how many frogs in each Ivermectin treatment were still alive then to be exposed to Bd? The timing aspect of the experimental design is very unclear in the current draft and made it difficult to interpret your results.

L219 and elsewhere – Include a space between digits and units.

L221 – What do you mean by “experienced a two-week infection period”? Were the frogs exposed to Bd once or repeatedly exposed the same dose of Bd throughout this period? If repeatedly, how often?

L224 – When were the animals euthanized relative to the Bd exposure?

L235 – Should say “euthanized”.

L236 – Remove comma between “experiment” and “died”.

L237 – Remove “during this study”.

L237-238 – Indicate earlier than no animals needed to be euthanized due to being unhealthy, then you can reduce your description of what you would have done if they had been.

L242 – When were these tissues collected, relative to either Bd or Ivermectin exposure?

L249 – Specify Ivermectin treatment.

L250 – Remove “on” before “growth rate”.

Figure 2 – I’m confused by the log curves. These seem to be fitting the raw data, not log-transformed data.

6. PLOS authors have the option to publish the peer review history of their article (what does this mean?). If published, this will include your full peer review and any attached files.

Reviewer #1: No

Reviewer #2: No

---

## [Author Response · Author response to Decision Letter 0]

14 Sep 2021

9/1/2021

Dear Dr. Dillman,

Thank you to both the Journal Staff and the Reviewers for such thorough and thoughtful comments on our manuscript. We appreciate the opportunity to revise our work in light of the feedback received. They greatly improved the overall quality of our work. Our responses to all individual points are made below in bold, italic text. We included the newly revised text where appropriate in response to better help individual reviewers see the changes made. We look forward to feedback on this revised manuscript.

Sincerely on behalf of all authors,

Jeffrey M. Grim

Journal Requirements:

https://journals.plos.org/plosone/s/file?id=ba62/PLOSOne_formatting_sample_title_authors_affiliations.pdf/

We have made our best effort to ensure that our revised submission satisfies PLOS ONE’s style requirements and apologize for any oversights that we may have made.

None of our references have been retracted, and some references were added in response to the Reviewer’s suggestions.

3. In your Methods section, please provide additional location information of the collection sites, including geographic coordinates for the data set if available. 

We have updated the Methods section to now read: 

“Eggs were collected from wild Cuban tree frogs (Osteopilus septentrionalis) by deploying plastic breeding refugia in local Tampa, FL, USA habitats (27.9506� N, 82.4572� W).”

4. In your Methods section, please provide additional information regarding the permits you obtained for the work. Please ensure you have included the full name of the authority that approved the collection sites access and, if no permits were required, a brief statement explaining why. 

We have updated the Methods section to now read:

No permits were required for egg collection as Cuban tree frogs are an invasive species in FL and all collections were done on private land with owner’s permission. No additional collecting permits were required.

Thank you for pointing this out. Funding from the University of Tampa (Faculty Development Dana and Delo Grants and Biology Department Student Research Funds) are given award numbers like larger, external grants. Consequently, all available grant numbers have been provided. 

We would like to acknowledge C. Nordheim and N. Laggan for their research assistance. Funding was provided by the National Science Foundation [IOS- 1754862 (TAM)] and the University of Tampa [Faculty Development Dana and Delo Grants (JMG and TAM) and Biology Student Research Funds (JMG and TAM)].

Funding was provided by the National Science Foundation [IOS- 1754862 (TAM)] and the University of Tampa [Faculty Development Dana and Delo Grants (JMG and TAM) and Biology Student Research Funds (JMG and TAM)].

We have removed the above Funding Statement from the Acknowledgments section of the manuscript. Please update the Funding Statement on the online form as:

Funding was provided to TAM (IOS award- 1754862) by the United States National Science Foundation (https://www.nsf.gov/div/index.jsp?div=IOS), and the Faculty Development Dana and/or Delo Grants and Biology Department Student Research Funds were provided to JMG and TAM by the University of Tampa (www.ut.edu). The funders had no role in study design, data collection and analysis, decision to publish, or preparation of the manuscript. 

The text has been edited to read: “All animal collections and procedures were approved in writing by the University of Tampa Institutional Animal Care and Use Committee (IACUC# 2018-12).”

The text has been edited to read: S1 Raw Data: These are the raw data sets used for all statistical analyses and figures.

 Reviewer #1: This is a well-written manuscript presenting the results of a long-term study on the effects of Ivermectin use in an amphibian species. The results are clearly presented with appropriate statistical analyses. R2 values should be added to Fig. 4. The role of the spleen in Bd infection dynamics is interesting. While the spleen size did not scale with body size, unlike the other organs (heart, liver), spleen size was correlated with Bd load. The Figure four results showing an interaction between Ivermectin exposure and spleen size and their impact on Batrachochytrium dendrobatidis (Bd) are fascinating, but the mechanism remains unclear. The authors may want to discuss additional studies to help clarify these findings, such as monitoring frogs post-treatment for Ivermectin half-life and clearance from the hosts. There are interesting associations with antibiotic use and growth and body mass index in humans during development. This may also be worthwhile to include in the discussion (e.g., doi: 10.1038/s41467-020-20495-4.). I personally suspect that the drug may have altered the microbiome of the amphibians which subsequently impacted immune function or directly impacts Bd resistance. Overall, this is an exciting study that has very careful conclusions that do not over-reach. The authors should be commended for their persistence at carrying out a long laboratory study looking for long-term effects.

R2 values should be added to Fig. 4.

Thank you for the suggestion. These have now been added.

The authors may want to discuss additional studies to help clarify these findings, such as monitoring frogs post-treatment for Ivermectin half-life and clearance from the hosts.

To our knowledge there are no Ivermectin half-life data (or clearance rate information) for Cuban tree frogs. You raise an important point. Future work would benefit from quantifying this information. The text now reads:

“While we acknowledge the restraints of lab resources and time, our data indicate the time frame considered for short and long term studies should be evaluated in the context of life history stage and overall lifespan of the organism, as well as species-specific drug half-lives.”

There are interesting associations with antibiotic use and growth and body mass index in humans during development. This may also be worthwhile to include in the discussion (e.g., doi: 10.1038/s41467-020-20495-4.).

Thank you for pointing out this interesting paper. Coupled with the results of our current study, the above paper certainly reinforces the idea that early pharmaceutical interventions can impact long term growth. This citation has been added and the concluding paragraph of now partially reads: 

“More recently, research has shown the potential efficacy of Ivermectin as therapuetic for SARS-CoV-2 in vitro (28), as a potential complimentary pharmaceutical treatment for combating malaria (29, 30), and a recent meta-analysis provides promising evidence supporting the removal of the contraindications for children <15 kg (31). However a growing literature suggests neonatal and early-childhood antibiotics impact child growth, BMI, and gut microbome in age-at-dose and sex-specific ways (e.g., (32)), indicating early pharmaceutical interventions can have long-term implications on organism growth. While we are certainly not advocating that Ivermectin use be eliminated, we want to make sure that it is clear that our results indicate there may similarly be long-term impacts of Ivermectin exposure in early developmental stages.”

I personally suspect that the drug may have altered the microbiome of the amphibians which subsequently impacted immune function or directly impacts Bd resistance.

This is an excellent suggestion and while not quantified in the current study, it would great to test these hypotheses in future work. 

Reviewer #2: General comments:

In the manuscript “Early-life exposure to Ivermectin alters long-term growth and disease susceptibility”, the authors assess the effects of a single exposure to Ivermectin on amphibian growth ad susceptibility to the pathogenic fungus Batrachochytrium dendrobatidis (Bd). Although I think these results warrant eventual publication, I think the manuscript needs to be improved in two main ways (in addition to minor adjustments elsewhere).

First, throughout the introduction and discussion, the authors highlight the potential application of studying long-term effects of Ivermectin exposure in amphibians to those effects in humans. Without heavy citation, I think that this is a leap. I think the authors’ efforts would be better directed towards focusing on the effects of Ivermectin on amphibians (which they do – I’d just like to see this be highlighted to a greater extent than a potential link to humans).

Thank you for comments on this. We worked hard to balance and respect amphibians and the implications of our work on this group singularly, while also recognizing the potential value this relatively novel model system could have to exploring the long-term implications of pesticide exposure to animals more broadly. Given the juxtaposition between yourself and Reviewer 1 (who suggested even more human tie-in) on our efforts, we have decided to leave these human connections with some edits.

Second, the experimental design and timeline require clarification. I highlight in line-by-line comments below specific areas of confusion that I had while reading. I assessed the results and discussion based on what I thought the authors did, but I’d like to see if I should revise my assessment once the methods are clarified.

Your suggestions related to the experimental design and timeline were excellent and improved the manuscript. We have made edits to the manuscript in several places described below under ‘Specific Comments’ in order to clarify the experimental design and timeline. 

Specific comments:

L87 – I think what you’re trying to get at here is that exposure to other chemicals, such as pesticides, can have long-lasting effects in amphibians, and, therefore, it is possible for Ivermectin to have long-lasting effects as well, but you could make this connection clearer.

You are correct and we have updated the text to be clearer about this potential connection. It now reads: 

“Early life exposure to pesticides is known to have long-lasting and often unpredictable impacts on amphibians, including increased mortality and disease risk into adulthood (14). Therefore, it seems plausible that Ivermectin exposure may similarly impact health.”

L91-94 – I don’t think you need to push the potential connection to human health as hard as you do here and throughout the manuscript. As you indicate, amphibians are important in their own right.

Thank you for comments on this. We worked hard to balance and respect amphibians and the implications of our work on this group singularly, while also recognizing the potential value this relatively novel model system could have to exploring the long-term implications of pesticide exposure to animals more broadly. Given the juxtaposition between yourself and Reviewer 1 (who suggested even more human tie-in) on our efforts, we have decided to leave these human connections with some edits. 

L97 – Please give some indication of survival rates in the different treatments.

Survival rates were 45% and 55% for control and Ivermectin-dosed groups, respectively. Thes data have been added to the Results and the text now reads:

“There was no effect of Ivermectin treatment on mortality (X21 = 0.904, p = 0.34; survival rates of 45% and 55% in control and Ivermectin-dosed groups, respectively) and there was no difference in frog mass at one-year post exposure (average mass � SEM: control: 0.668 � 0.02; Ivermectin: 0.674 � 0.03; X21 = 0.02, p = 0.877; Fig 1).”

L99 (and elsewhere) – Sometimes you capitalize “Ivermectin” and sometimes you don’t. Be consistent.

Thank for your attention to detail with this. We have updated all references to the “Ivermectin”.

L101 – Specify that this result is relative to those not exposed to Ivermectin.

We have updated the manuscript so that it now reads:

“Ivermectin had reduced growth rate (2.5 years of growth: X21 = 7.78, p = 0.005; Fig 2) and lower change in mass (X21 = 5.36, p = 0.02) at the end of the experiment, relative to unexposed animals.

L102 – I think this is the first time you mention “metabolically-expensive organs”. It would be beneficial to explain sooner why you chose to measure the liver, heart, and spleen.

We have edited the ‘Tissue Collection’ section of the Methods and combined it with the Bd exposure section to now read:

Bd exposure and tissue sampling 

To evaluate the effects of Ivermectin exposure on Bd susceptibility, all adult frogs from both the control and Ivermectin-exposed groups were dosed with 1 mL of the Bd+ stock during the final two weeks of the overall 2.5 experimental period. Bd+ stock was squirted directly onto the dorsal surface of each frog. The extra liquid was allowed to collect on the damp paper towels in the 1 L plastic container to promote infection establishment. Following the singular Bd+ dose, the frogs experienced a two-week infection period, during which container changes occurred weekly, they were fed calcium dusted crickets ad libitum, and they were maintained at 18 °C to promote Bd growth. Animals were euthanatized immediately at the end of this two-week infection period with an overdose of topical benzocaine and they were pithed post mortem. A ventral skin sample was collected from each frog (1 cm2) for quantitative PCR to determine Bd infection load (See 31 for the qPCR protocol used). In order to assess the potential effects of Ivermectin treatment on the partitioning of resources between whole animal growth and the growth of individual organs, we also sampled the metabolically-expensive heart, liver, and spleen from each euthanized individual. Organs were collected and rinsed in a 0.7% saline bath, and were weighed immediately. All animal procedures were approved by the University of Tampa IACUC (2018-12). 

L108-109, and 118-120 – Indicate what error bars represent in all figure captions.

Figure captions (general) – I think it would be helpful to have all captions highlight a specific result, as the caption for Figure 4 does (there it’s the interactive effect of Ivermectin exposure and spleen size on Bd load).

We have edited the Figure captions to now read: 

“Fig. 1. The effects of early life exposure of Ivermectin on mass of Cuban tree frogs (Osteopilus septentrionalis) one year after metamorphosis. No significant difference in mass was observed. Shown are means ± SEM. 

Fig. 2. The effects of early life exposure of Ivermectin on the long-term growth of Cuban tree frogs (Osteopilus septentrionalis). Individuals exposed to Ivermectin had slower growth rates and weighed less, relative to controls. Shown are means ± SEM.

Fig. 3. The effects of early life exposure of Ivermectin on the disease susceptibility in Cuban tree frogs (Osteopilus septentrionalis) to Batrachochytrium dendrobatidis (Bd; Bd load is reported in zoospore genome equivalents. Individuals exposed to Ivermectin were less susceptible to Bd infection compared to controls. Shown are means ± SEM.

Fig. 4. There was an interaction between Ivermectin exposure and spleen size and their impact on Batrachochytrium dendrobatidis (Bd) load in Cuban tree frogs (Osteopilus septentrionalis).“

L115-117 – Run-on sentence.

The manuscript now reads: 

“There was a negative correlation between spleen size and Bd load for individuals exposed to Ivermectin, while for the control individuals the correlation was positive (Fig 4).”

L120 – Should say “zoospore genome equivalents).”

Thank you for the correction.

L128-130 – Needs citation(s).

We have added citations here, including a more recent review and two earlier studies that explored the safety of various Ivermectin doses and also dosage time frames.

L141-144 – I think it’s a pretty big leap from metamorphic frogs to human children here, particularly without citation.

We have removed the line here and it now reads: 

“Given that these individuals only experienced one dose, it is likey that our results underestimate the true impact of repeated Ivermecting exposure during early development.”

However, it is worth noting that Ivermectin utilization during mass drug administration targeting neglected tropical diseases should preclude administration to young children as there are existing contraindications for individuals < 15 kg. A recent review concluded that there were “limited but encouraging evidence that Ivermectin is safe and well-tolerated in small children weighing less than 15 kg.” … “Further investigation is warranted through well-designed clinical trials in children weighing less than 15 kg with the objective of optimizing dosing and characterizing the safety profile the prescribing restriction in young children can be lifted.” Also as pointed out by Reviewer #1, neonatal and early childhood antibiotic use may also impact long-term child growth, BMI, and gut microbiome in age-at-dose and sex-specific ways, thus it seem possible that other pharmaceutical interventions might also have similar effects.

Therefore, we have added text to the concluding paragraph of the discussion so that it now partially reads:

“More recently, research has shown the potential efficacy of Ivermectin as therapuetic for SARS-CoV-2 in vitro (28), as a potential complimentary pharmaceutical treatment for combating malaria (29, 30), and a recent meta-analysis provides promising evidence supporting the removal of the contraindications for children <15 kg (31). However a growing literature suggests neonatal and early-childhood antibiotics impact child growth, BMI, and gut microbome in age-at-dose and sex-specific ways (e.g., (32)), indicating early pharmaceutical interventions can have long-term implications on organism growth. While we are certainly not advocating that Ivermectin use be eliminated, we want to make sure that it is clear that our results indicate there may similarly be long-term impacts of Ivermectin exposure in early developmental stages.”

L149 – Specify Ivermectin treatment.

The text now reads:

“We found that while there was a positive correlation between frog mass and organ (liver and heart) mass, which was expected, there was no effect of Ivermectin treatment on the mass of these metabolically expensive organs.”

L149-152 – I’m having a hard time following this result. Is there a way to show it in figure form?

We have edited the manuscript in an effort to clarify this result. It now reads: 

“Organisms operate with limited energetic resources, and during development these resources must be differnetially allocated to growth, the development of crucial organs (e.g. heart and liver), and pathogen defense (e.g. immune response). We found that while there was a positive correlation between frog mass and organ (liver and heart) mass, which was expected, there was no effect of Ivermectin treatment on the mass of these metabolically expensive organs. Ivermectin exposure reduced whole body growth and growth rate when compared to control animals, but did not result in smaller hearts, livers, or spleens of these individuals. This indicates Ivermectin influences growth in organ-specific ways, as metabolically expensive (and important) organs likely received an increased allocation of resources relative to whole body growth following Ivermectin treatment. This disparate allocation is not surprising given that studies have shown that resources are shunted toward critical organs and tissues in vertebrates in tissue-specific ways and under resource-limiting conditions (22, 23).”

L150 – “Metabolically” is misspelled.

This has been corrected.

L159-160 – I think you’d benefit here from including results of prior studies regarding the relationship between spleen size and Bd load.

We added in appropriate citations and the manuscript now reads: “Interestingly, Bd exposure alone was found to have no impact on spleen size (Fites et al 2014 and Brannelly et al 2016), but frogs with an acquired immune response to Bd had increased splenocyte production (McMahon et al 2014).”

L178 – Remove “whole heartedly”.

This has been removed.

L186 – “Pharmaceutical” is misspelled.

Thank you for catching this mistake. It has been corrected.

L191 – Remove the comma between “Ivermectin” and “is”.

This change has been made.

L203 – Please explain the Ivermectin treatment in more detail. I see in L139 that there was a single Ivermectin exposure, but that should also be stated here. Additionally, were all frogs dosed with Ivermectin at the same absolute time (as opposed to the same amount of time between metamorphosis and exposure for each frog)? And if so, approximately how long after metamorphosis?

Egg clutches were collected at the same time and animals assigned to the groups were all approximately 4 weeks post metamorphosis. All animals were dosed at the same time.

This now reads: 

Metamorphic frogs (~ 4 weeks post metamorphosis) were divided evenly and randomly between control and Ivermectin-dosed groups (n = 20/treatment). They were all given a single dose orally of either ASW (control) or 1% Ivermectin (2 mg/kg body mass/treatment) at the same time. 

L218 – Relative to Ivermectin exposure, when were frogs exposed to Bd? After reading through a couple times, I’m guessing this was in the final two weeks of the experiment? And If so, how many frogs in each Ivermectin treatment were still alive then to be exposed to Bd? The timing aspect of the experimental design is very unclear in the current draft and made it difficult to interpret your results.

The text has been edited in two places to more clearly communicate the timeline of our work. Briefly, the entire experiment lasted 2.5 yrs beginning with the metamorphic phase and ended with a 2 week Bd infection period. The text now reads:

“Mortality was checked daily, container changes occurred weekly, and the frogs were weighed every week through their exponential growth phase (approximately 2.5 years), including during the final two week Bd infection period for both control and Ivermectin exposed groups.”

“To evaluate the effects of Ivermectin exposure on Bd susceptibility, all surviving adult frogs in both control (n=9) and Ivermectin-exposed (n=11) groups were dosed with 1 mL of the Bd+ stock during the final two weeks of the overall 2.5 year experimental period. Bd+ stock was squirted directly onto the dorsal surface of each frog.”

L219 and elsewhere – Include a space between digits and units.

The text has been edited accordingly.

L221 – What do you mean by “experienced a two-week infection period”? Were the frogs exposed to Bd once or repeatedly exposed the same dose of Bd throughout this period? If repeatedly, how often?

The text now reads”

“Following the singular Bd+ dose, the frogs experienced a two-week infection period, during which container changes occurred weekly, they were fed calcium dusted crickets ad libitum, and they were maintained at 18 °C to promote Bd growth.”

L224 – When were the animals euthanized relative to the Bd exposure?

The text now reads:

“Animals were euthanatized immediately at the end of this two-week infection period with an overdose of topical benzocaine and they were pithed post mortem.”

L235 – Should say “euthanized”.

Thank you for correcting this typo.

L236 – Remove comma between “experiment” and “died”.

The manuscript has now been changed.

L237 – Remove “during this study”.

The manuscript has now been changed.

L237-238 – Indicate earlier than no animals needed to be euthanized due to being unhealthy, then you can reduce your description of what you would have done if they had been.

We appreciate your suggestion here, however this section was added at the request of the journal and therefore we have left it in tact.

L242 – When were these tissues collected, relative to either Bd or Ivermectin exposure?

In an effort to more clearly represent the experimental time line, we consolidated sections of the Methods which now reads: 

“Bd exposure and tissue sampling 

To evaluate the effects of Ivermectin exposure on Bd susceptibility, all surviving adult frogs in both control (n=9) and Ivermectin-exposed (n=11) groups were dosed with 1 mL of the Bd+ stock during the final two weeks of the overall 2.5 year experimental period. Bd+ stock was squirted directly onto the dorsal surface of each frog. The extra liquid was allowed to collect on the damp paper towels in the 1 L plastic container to promote infection establishment. Following the singular Bd+ dose, the frogs experienced a two-week infection period, during which container changes occurred weekly, they were fed calcium dusted crickets ad libitum, and they were maintained at 18 °C to promote Bd growth. Animals were euthanatized immediately at the end of this two-week infection period with an overdose of topical benzocaine and they were pithed post mortem. A ventral skin sample was collected from each frog (1 cm2) for quantitative PCR to determine Bd infection load (See 34 for the qPCR protocol used). In order to assess the potential effects of Ivermectin treatment on the partitioning of resources between whole animal growth and the growth of individual organs, we also sampled the metabolically-expensive heart, liver, and spleen from each euthanized individual. Organs were collected and rinsed in a 0.7% saline bath, and were weighed immediately. All animal procedures were approved by the University of Tampa IACUC (#2018-12). 

L249 – Specify Ivermectin treatment.

Thank you. This change has been made.

L250 – Remove “on” before “growth rate”.

Thank you. This change has been made.

Thank you for noting this. We adjusted the figure to have linear lines and added R2 values for consistency with Figure 4.

---

## [Editor Report · Decision Letter 1]

22 Sep 2021

Early-life exposure to Ivermectin alters long-term growth and disease susceptibility

PONE-D-20-33880R1

Dear Dr. Grim,

We’re pleased to inform you that your manuscript has been judged scientifically suitable for publication and will be formally accepted for publication once it meets all outstanding technical requirements.

Kind regards,

Adler R. Dillman, Ph.D.

Academic Editor

PLOS ONE

Additional Editor Comments (optional):

Thank you for your patience during this unusually lengthy review period. The revised manuscript addresses reviewer concerns and makes important contributions to the field.
---

## [Editor Report · Acceptance letter]

5 Oct 2021

PONE-D-20-33880R1 

Early-life exposure to Ivermectin alters long-term growth and disease susceptibility 

Dear Dr. Grim:

I'm pleased to inform you that your manuscript has been deemed suitable for publication in PLOS ONE. Congratulations! Your manuscript is now with our production department. 

Kind regards, 

on behalf of

Dr. Adler R. Dillman 

Academic Editor

PLOS ONE